# Glutamine supplementation improves the efficacy of miltefosine treatment for visceral leishmaniasis

**Carolina Ferreira[1,2], Inês Mesquita[1,2], Ana Margarida Barbosa[1,2], Nuno Sampaio Osório[1,2], Egídio Torrado[1,2], Charles-Joly Beauparlant[3,4], Arnaud Droit[3,4], Cristina Cunha[1,2], Agostinho Carvalho[1,2], Bhaskar Saha[5,6], Jerôme Estaquier[4,7], Ricardo Silvestre[1,2]***

**1** Life and Health Sciences Research Institute (ICVS), School of Medicine, University of Minho, Braga, Portugal, **2** ICVS/3B's-PT Government Associate Laboratory, Braga/Guimarães, Portugal, **3** Département de Médecine Moléculaire—Université Laval, Faculté de médecine, Québec, Canada, **4** Centre de Recherche du CHU de Québec—Université Laval, Québec, Canada, **5** National Centre for Cell Science, Pune, India, **6** Trident Academy of Creative Technology, Bhubaneswar, Odisha, India, **7** INSERM U1124, Université de Paris, Paris, France

* ricardosilvestre@med.uminho.pt

**Data Availability Statement:** The RNAseq data generated during this study are available at Gene

## Abstract

### Background

The disturbance of host metabolic pathways by *Leishmania* parasites has crucial consequences for the activation status of immune cells and the outcome of infection. Glutamine has been described as an immunomodulatory amino acid, yet its role during *Leishmania* infection is still unknown.

### Methods

We performed transcriptomics in uninfected and *L. donovani*-infected macrophages 6 hours post-infection. Glutamine quantification by HPLC was assessed in the supernatant of macrophages throughout the infection course. For experimental *L. donovani* infections, mice were infected with $1.0 \times 10^8$ stationary *L. donovani* promastigotes. Glutaminase (GLS) chemical inhibition was performed using BPTES and glutamine was administered throughout infection. For combined therapy experiment, a daily administration of miltefosine and glutamine was performed by oral gavage. Parasite burden was determined using a Taqman-based assay. Immune cell phenotyping and cytotoxicity were performed in splenic cells using flow cytometry.

### Findings

We show that glutamine is essential for the control of *L. donovani* infection. Transcriptomic analysis of *L. donovani*-infected macrophages demonstrated an upregulation of genes involved in glutamine metabolism. Pharmacological inhibition of glutaminolysis significantly increased the susceptibility to infection, accompanied by an increased recruitment of anti-inflammatory myeloid cells and impaired T cell responses. Remarkably, the supplementation of glutamine to mice infected with *L. donovani* during miltefosine treatment potentiates

Expression Omnibus (GEO) under the accession number GSE145136.

**Funding:** This work was supported by the Northern Portugal Regional Operational Programme (NORTE 2020), under the Portugal 2020 Partnership Agreement, through the European Regional Development Fund (FEDER) (NORTE-01-0145-FEDER-000013) and the Fundação para a Ciência e Tecnologia (FCT) (contracts PD/BDE/127830/2016 to CF, SFRH/BD/120127/2016 to IM, SFRH/BD/120371/2016 to AMB, IF/00474/2014 to NSO, IF/01390/2014 to ET, CEECIND/03628/2017 to AC, CEECIND/04600/2017 to CC and IF/00021/2014 to RS), and Infect-Era Net (project INLEISH). JE also thanks the Canada Research Chair program for financial assistance. The funders had no role in study design, data collection and analysis, decision to publish, or preparation of the manuscript.

**Competing interests:** The authors have declared that no competing interests exist.

parasite clearance through the development of a more effective anti-*Leishmania* adaptive immune response.

## Conclusions

Our data indicates that dietary glutamine supplementation may act as a promising adjuvant for the treatment of visceral leishmaniasis.

### Author summary

Visceral leishmaniasis is a life threatening neglected tropical disease affecting around 500,000 and killing 50,000 individuals a year. Despite its obligatory dependence on host cell metabolism and the lack of effective, non-toxic, orally bioavailable anti-leishmanial drugs, *Leishmania*-perturbed host cell metabolomes and its relation to anti-leishmanial chemotherapy remains unexplored. Transcriptomic analysis performed on *L. donovani*-infected macrophages identified patterns of gene expression associated with glutamine metabolism. *In vitro* and *in vivo* pharmacological inhibition of glutaminase (GLS), which catalyzes the first reaction in the primary pathway for the catabolism of glutamine, significantly increased the susceptibility to infection demonstrating the role of glutamine metabolism to *L. donovani* infection. More importantly, we demonstrated that glutamine supplementation during miltefosine treatment potentiates *L. donovani* clearance through the development of a more effective anti-*Leishmania* innate and adaptive immune response. Overall, our work demonstrated that glutamine-miltefosine synergy is a novel combined host- and pathogen-directed treatment for combating visceral leishmaniasis.

## Introduction

Host-parasite interactions are multifaceted involving several factors that allow pathogens to evade the immune system and replicate [1,2]. In response to pathogens, immune cells undergo a wide range of phenotypical alterations, particularly in their metabolism, to optimize their immune effector functions. It is now well established that the metabolic reprogramming of immune cells is extremely important for the development of an optimal immune response [3–5]. *Leishmania* perturbs the metabolic pathways in immune cells causing long-term functional changes to manipulate the infection outcome [2,6]. *Leishmania* parasites manipulate macrophage metabolism switching from an early glycolytic phenotype to an oxidative phosphorylation profile, in a process dependent on the energy sensors SIRT1 and LKB1/AMPK [7]. Both AMPK and SIRT1 have been reported to regulate the consumption of glutamine, which is one of the most abundant amino acids in the plasma contributing to the anaplerotic reactions of the TCA cycle, supporting ATP production and fatty acid synthesis [8]. Once transported by the neutral amino acid transporter B(0) (SLC1A5) into the cell, glutamine sequentially undergoes glutaminolysis by glutaminase (GLS) to glutamic acid, which is converted by glutamate dehydrogenase (GDH) to α-ketoglutarate (α-KG), a substrate for TCA cycle, that is required for the metabolism of C2 units.

Recent reports have highlighted the impact of glutamine metabolism in macrophage biology. Increased glutamine metabolism was held accountable for an epigenetic regulation responsible for the M2-like macrophage polarization observed in response to IL-4 stimulation [9,10]. Glutaminolysis was also found to be essential for the increased cytokine production

and epigenetic changes observed in β-glucan-induced trained immunity [11]. In the context of infection, although less understood, glutamine was shown to act as a major energy source for the survival of HIV-infected macrophages and to impact cytokine response during *M. tuberculosis* infection [12,13]. Therefore, changes in glutamine metabolism on macrophages could impact their capacity to provide an adequate support for eliciting of an efficient adaptive immune response. In fact, glutamine metabolism is also essential for T cell activation, proliferation and cytokine production [14,15]. Yet, the role of glutamine and glutaminolysis in the context of *Leishmania* infection remained elusive.

Herein, we demonstrate that *L. donovani*-infected macrophages display an upregulation of genes involved in glutamine regulation along with an increased glutamine consumption. *In vivo* blocking of glutamine conversion into glutamate using bis-2-(5-phenylacetamido-1,2,4-thiadiazol-2-yl) ethyl sulfide (BPTES), an inhibitor of GLS, resulted in the accumulation of splenic monocytes displaying an anti-inflammatory profile. Concomitantly, we found a reduction in both splenic CD4 and CD8 T cell numbers and a decrease in IFN-γ and TNF-α, two major effector cytokines involved in parasite clearance. Importantly, this immune profile was associated with an increase in parasite load. Finally, glutamine supplementation synergizes with miltefosine anti-parasitic activity improving macrophage activation and the frequency of CD4 and CD8 T cells-expressing IFN-γ/TNF-α. Taken together, our results demonstrate that dietary glutamine supplementation may represent a previously unappreciated adjunct therapy for the control of visceral leishmaniasis (VL) for which the repertoire of drugs is severely limited.

## Material and methods

### Ethics statement

The experiments were conducted with the approval of the UMinho Ethical Committee (process no. SECVS 074/2016) and complied with the guidelines of the Committee and National Council of Ethics for the Life Sciences (CNECV). CF and RS has an accreditation for animal research given from Portuguese Veterinary Direction (Ministerial Directive 1005/92).

### Mice

C57BL/6 mice were purchased from Charles River Laboratories and maintained in accredited animal facilities at the Life and Health Sciences Research Institute (ICVS). Both males and females with 8–12 weeks were used.

### Parasite culture and staining

Cloned lines of virulent *Leishmania donovani* (MHOM/IN/82/Patna1) were maintained with weekly sub-passages at 27ºC in complete RPMI 1640 medium, supplemented with 10% heat-inactivated fetal bovine serum, 2 mM L-glutamine, 100 U/ml penicillin plus 100 mg/ml streptomycin and 20 mM HEPES buffer (ThermoFisher).

### Macrophage culture and *in vitro* infections

For peritoneal macrophages, peritoneal exudate was recovered after injection of ice-cold PBS in the peritoneal cavity. Cells were washed, resuspended in complete RPMI and plated at a density of 1 x $10^6$/ml. Non-adherent cells were removed through washing 4 hours after plating. Bone marrow-derived macrophages (BMDMs) were differentiated using recombinant M-CSF (Peprotech) as previously described [16]. Macrophages were infected with *L. donovani* promastigotes at a 1:10 ratio. After 4 hours of incubation, non-phagocytosed parasites were removed through washing and cells were recovered at the designated time points. Two days

post-infection, macrophages were treated with bis-2-(5-phenylacetamido-1,2,4-thiadiazol-2-yl) ethyl sulfide (BPTES; 10 μM) (Sigma Aldrich) or left untreated as control until day five post-infection.

### *In vitro* cytotoxicity assays

The *in vitro* cytotoxicity assay was performed by flow cytometry using 7-AAD staining (Invitrogen, Life Technologies) as previously reported [17,18]. BMDMs-infected 24 hours with *L. donovani* were used as target cells. Nonadherent splenocytes from one-month *L. donovani*-infected mice were stimulated with SLA (10μg/ml) and IL-2 (20ng/ml) for 3 days, after which they were recovered and used in the cytotoxic assay. Briefly, effector cells were incubated for 48 hours with previously *L. donovani*-infected BMDMs at different E:T cell ratios (2:1; 5:1 and 10:1), in the presence of normal levels of L-glutamine (2mM; CTRL), high levels of glutamine (5 mM; + L-glut) and without glutamine (- L-glut).

### Parasite viability *in vitro*

Five days post-infection, culture media was renewed (200 μl of fresh media per well). The 96-well plates were then tightly sealed and incubated at 27ºC for 7 days, to allow the conversion of intracellular amastigotes in promastigotes. The parasites were then fixed with 2% PFA and counted in a Neubauer chamber.

### Experimental *Leishmania* infection

Mice were infected with 1.0 x $10^8$ (intraperitoneal route) stationary *L. donovani* promastigotes and weight and well-being was monitored during the infection. Intraperitoneal inoculation was used given that it results in a higher homogeneity of infections between animals [19,20]. For GLS inhibition experiment early on infection, BPTES was administered via intraperitoneal route, at days 2, 4, 6, 8 and 10 post-infection (12.5 mg/kg). Control animals were equally injected with vehicle (sterile corn oil and 5% DMSO). Glutamine administration was performed by oral gavage (500 mg/kg/day) throughout infection. The dose was chosen based on the previous reported data in humans [21] and according with the daily intake of glutamine in previous studies with rodents [22,23]. Experimental groups were kept in separated cages. Mice were euthanized 12 days post-infection. For combined therapy experiment using miltefosine and glutamine, L-glutamine (500 mg/kg/day) was administered by oral gavage daily (starting at day 18 post-infection). Miltefosine (25 mg/kg) was administered daily by oral gavage (starting at day 25 post-infection). Mice were euthanized one-month post-infection. The splenocytes were prepared for flow cytometry analysis and DNA extraction. DNA was extracted using the phenol-chloroform-isoamyl alcohol method. Parasite burden was assessed, using a TaqMan-based qPCR assay for detection and quantitation of *L. donovani* kinetoplastid DNA [24].

### RNA sequencing

RNA sequencing was performed in uninfected and *L. donovani* infected bone-marrow macrophages at 6 hours post-infection. The NEBNext Ultra II directional RNA library prep kit for Illumina (New Englands Biolabs Inc., Ipswich, MA, USA) was used to prepare total RNA sequencing libraries, according to manufacturer's instruction. Briefly, 100 ng of total RNA was used. Ribosomal RNA (rRNA) including both cytoplasmic and mitochondrial rRNA was removed using RNaseH-based method (NEBNext rRNA depletion kit; New Englands Biolabs Inc., Ipswich, MA, USA). Following purification with Agencourt RNAClean XP beads (Beckman Coutler, Missisauga, Ontario, Canada), the RNA was fragmented using divalent cations

under elevated temperature. The fragmented RNA was used as a template for cDNA synthesis by reverse transcriptase with random primers. The specificity of the strand was obtained by replacing the dTTP with the dUTP. This cDNA was subsequently converted to double-stranded DNA and end-repaired. Ligation of adaptors was followed by a purification step with AxyPrep Mag PCR Clean-up kit (Axygen, Big Flats, NY, USA), by an excision of the strands containing the dUTPs and finally, by a PCR enrichment step of 11 cycles to incorporate specific indexed adapters for the multiplexing. The quality of final amplified libraries was examined with a DNA screentape D1000 on a TapeStation 2200 and the quantification was done on the QBit 3.0 fluorometer (ThermoFisher Scientific, Canada). Subsequently, total RNA-seq libraries with unique index were pooled together in equimolar ratio and sequenced for paired-end 125 pb sequencing using one lane of a high output flowcell on an HiSeq 2500 V4 system at the Next-Generation Sequencing Plateform, Genomics Center, CHU de Québec-Université Laval Research Center, Québec City, Canada. Reads were trimmed using Trimmomatic v0.36 with the following options: TRAILING:30, SLIDINGWINDOW:4:20 and MINLEN:30. All other options used the default values. Quality check was performed on raw and trimmed data to ensure the quality of the reads using FastQC v0.11.5 and MultiQC v1.5. The quantification was performed with Kallisto v0.44. Differential expression analysis was performed in R v3.5.0 using the DESeq2 v1.20.0.

## Flow cytometry

The spleens were minced and forced through a 70-μm cell strainer (Corning Inc.) and red blood cells were lysed with ACK lysis buffer (0.15 M $NH_4Cl$, 10 Mm KHCO3 and 0.1 mM EDTA). The anti-mouse monoclonal antibodies used to perform this study were purchased to BioLegend (CA, USA) and shown in Supplementary S1 Table. Samples were acquired on a LSRII flow cytometer (BD Biosciences) and data analyzed using FlowJo software (FlowJo LLC).

## Glutamine quantification by High Performance liquid chromatography (HPLC)

Precolumn derivatization method using ortho-phthalaldehyde (OPA with methanol ≥99.9%, potassium borate 1M pH = 9.5, and 2-mercaptoethanol ≥99.0%) 1:5 (Sigma Aldrich) was used to detect L-glutamine by a Gilson UV/vis_155 detector (338nm). Culture supernatants were filtered by Acrodisc 13mm syringe filters with 0.2μm supor membrane (Pall Corporations) previously to analysis. Inorganic mobile phase A, pH = 7.8, was composed by $Na_2HPO_4.2H_2O$ 350 mM:propionic acid 250mM (1:1) mixture (Merck) with acetonitrile HPLC grade in water (10:2:13) and organic mobile phase B was composed by acetonitrile, methanol and water (3:3:4) (HPLC grade, HiPerSolv Chromanorm, VWR Chemicals). Standard solutions were prepared in MilliQ water (Millipore). A Gilson bomb system (Gilson) was used with a 40ºC Hi-Chrom C18 (model HI-5C18-250A) 5μm particles column (HiCrom). All data was analyzed in Gilson Uniprot Software, version 5.11.

## Quantitative PCR analysis

Total RNA was isolated from uninfected and *L. donovani* infected bone-marrow macrophages (during 6, 24 and 48 hours) with TRI reagent (Sigma), according to the manufacturer instructions. RNA concentration was determined by OD260 measurement using a NanoDrop spectrophotometer. Total RNA (10–250 ng) was reverse transcribed using the Xpert cDNA synthesis Mastermix (Grisp). Real-Time quantitative PCR (qRT-PCR) reactions were run for each sample on a Bio-Rad CFX96 Real-Time System C1000 Thermal Cycler (Bio-rad). Primer

sequences were obtained from Integrated DNA technologies (USA). The RT product was expanded using the SensiFAST SYBR Hi-ROX kit (Bioline) and the results were normalized to the expression of the housekeeping gene *28s*. After amplification, cycle threshold-values (Ct-values) were calculated for all samples and gene expression changes were analyzed in the CFX Manager Software (Bio-Rad). The qPCR primers used for this study are listed in Supplementary S2 Table.

### Statistical analysis

Statistical analyses as described in figure legends were performed in GraphPad Prism 6 software. A two-way analysis of variance (two-Way ANOVA) followed by a Sidak's post-test, Mann-Whitney test of variance and Kruskal-Wallis non-parametric test followed by a Dunn's post-test were performed accordingly with the correspondent experimental design. ANOVA assumptions were analyzed through normality test (Shapiro-Wilk test, p>0,05) and homogeneity of variances evaluation. Statistically significant values are as follows: $^{*}$p < 0.05, $^{**}$p < 0.01, $^{***}$p < 0.001.

## Results

### Glutamine metabolism is upregulated upon *L. donovani* infection

Transcriptomic analysis of *L. donovani*-infected macrophages at 6h post-infection identified patterns of gene expression associated with glutamine metabolism (Fig 1A). *L. donovani* infection led to a rapid and transient induction of genes involved in glutamine utilizing reactions (Fig 1A and 1B), while downregulating several TCA-related genes (Fig 1A). Specifically, we observed a transient increase in *Glul*, *Gfpt2*, *Gclm*, *Gclc* and *Gpt2* mRNA expression at 6h (Fig 1B). A second wave of transcription induction was observed at 48h post-infection and included *Glud1* and *Gls* involved in the transformation of glutamine to glutamate (glutaminolysis-related enzymes) (Fig 1B). No significant changes were observed with *Got2* and the glutamine associated transporters *Slc1a5* and *Slc7a5*. To understand if the increased expression of glutaminolysis-related genes was correlated with an increased glutamine consumption during *L. donovani* infection, we quantified glutamine levels in the supernatant of uninfected and *L. donovani*-infected cells. We observed that glutamine consumption was increased in infected cells compared to uninfected macrophages (Fig 1C). To determine the impact of glutaminolysis impairment during *L. donovani* infection *in vitro*, we have pharmacologically inhibited GLS, using BPTES, 48 hours post-infection. We found that *in vitro* BPTES-treated macrophages display, on day 5 post-infection (120 hrs), a modest but significant increase in parasite burden (Fig 1D). These results show that glutamine metabolism is upregulated during *L- donovani* infection and plays a crucial role in the ability of macrophages to control parasite growth.

### Pharmacological inhibition of GLS increases susceptibility to *L. donovani* infection

As glutamine metabolism is upregulated during *L. donovani* infection, we started to examine the impact of glutamine metabolism in the antigen presentation capacity of infected macrophages by evaluating the cytotoxicity capacity of *L. donovani*-specific T cell during *in vitro* glutamine deprivation and supplementation. Spleen cells recovered on day 30 post-infection from *L. donovani*-infected mice were enriched in *Leishmania*-specific clones by the *ex vivo* stimulation with SLA and IL-2 for 3 days. Cells were then tested for their cytotoxic activity against *L. donovani*-infected BMDMs in the presence of normal (2 mM; CTRL) or higher levels of glutamine (5 mM; +L-glut) and in the absence of this amino acid (- L-glut). We found that the absence of glutamine led to a reduction of the capacity of *L. donovani* specific T cells to lyse

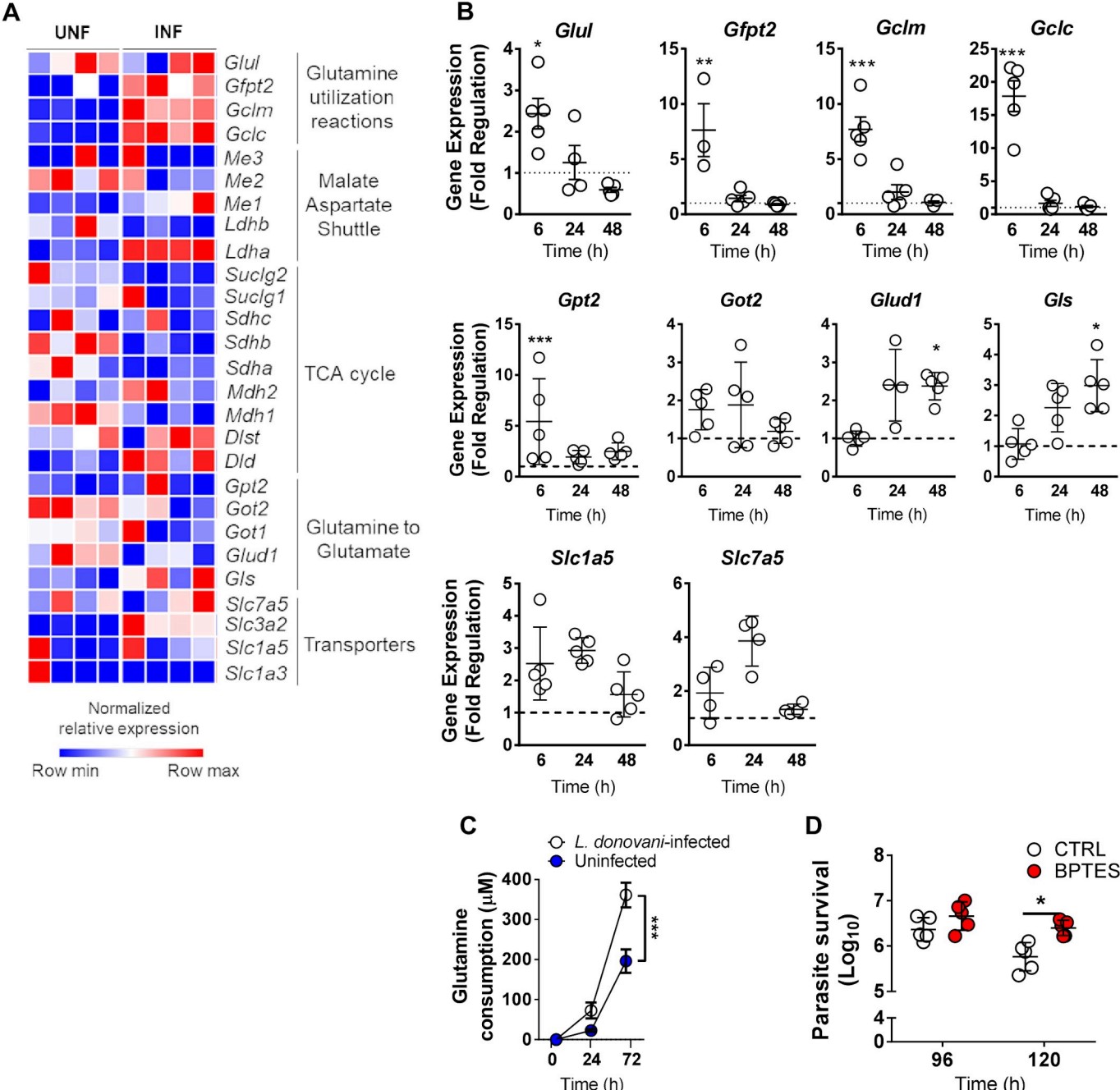

Fig 1. *L donovani* infection is associated with increased glutamine metabolism. (A) Heatmap of gene expression pattern of glutamine genes in uninfected or *L. donovani*-infected macrophages 6 hours post-infection using RNA sequencing analysis (n = 4); (B) mRNA expression of glutamine synthetase (*Glul*), glutamine-fructose-6-phosphate aminotransferase (*Gfpt2*), glutamate—cysteine ligase regulatory subunit (*Gclm*), glutamate-cysteine ligase catalytic subunit (*Gclc*), alanine aminotransferase 2 (*Gpt2*), aspartate aminotransferase (*Got2*), glutamate dehydrogenase 1 (*Glud1*), Glutaminase (*Gls*), neutral amino acid transporter B(0) (*Slc1a5*) and large neutral amino acids transporter small subunit 1 (*Slc7a5*) in murine macrophages infected with *L. donovani* for 6, 24 or 48 hours relative to uninfected cells using quantitative PCR analysis (n = 5). The interaction between the effects of infection and time in the transcription levels of *Glul* (($F_{(2,16)}$ = 7.191, $p$ = 0.0059), *Gfpt2* ($F_{(2,16)}$ = 9.376, $p$ = 0.0020), *Gclm* ($F_{(2,16)}$ = 24.87, $p<0.0001$) and *Gclc* ($F_{(2,16)}$ = 86.35, $p<0.0001$) was found significant while no major differences were found for the other evaluated enzymes and transporters. Simple main effect analysis showed that the transcription levels of *Gpt2* ($F_{(1,8)}$ = 19.92, $p$ = 0.0021), *Glud1* ($F_{(1,8)}$ = 13.35, $p$ = 0.0065) and *Gls* ($F_{(1,8)}$ = 8.443, $p$ = 0.0197) were found increased in infected cells compared to uninfected macrophages; (C) Glutamine consumption on the supernatants of in vitro uninfected murine macrophages before infection (0) and upon infection with *L. donovani* for 24 and 72 hrs (n = 5). A two-way ANOVA was used to evaluate glutamine consumption by uninfected and infected cells. The interaction between the effects of infection and time in glutamine consumption was significant ($F_{(2,16)}$ = 11,66, p = 0,0008). Simple main effect analysis showed that glutamine consumption was increased in infected cells compared to uninfected macrophages ($F_{(1,8)}$ = 15,75, $p$ = 0,0041), and that the timepoint post-infection was found to

be an important variable in this analysis (F(2,16 = 144,6, *p*<0,0001. (D) Bone marrow-derived macrophages were infected with *L. donovani* promastigotes (1:10 ratio) (n = 5). Pharmacological inhibition of glutaminase (GLS) was performed 48 hours post-infection using BPTES (10uM). Parasite viability was assessed at 96- and 120-hours post-infection. A Mann-Whitney t-test was used to evaluate parasite viability in the presence of BPTES. The presented data is representative of three independent experiments and values are expressed as mean ± SD. p<0.05; *p<0.01; ***p<0.001.

infected macrophages (45% lysis of target cells at 10:1 E:T ratio) compared to the cells cultured in the presence of normal levels of glutamine (60% lysis of target cells at 10:1 E:T ratio). Importantly, higher levels of glutamine enhanced the cytotoxicity reaching 75% of lysed macrophages (Fig 2A). These results support the importance of glutamine metabolism for *Leishmania* specificity of cytotoxic cells.

Next, we assessed the *in vivo* impact of a specific GLS inhibitor (BPTES, 12.5 mg/kg every two days) or providing glutamine supplementation at a daily dose of 500 mg/kg during the early steps of the infection (Fig 2B). GLS inhibition increased the recruitment of myeloid cells to the spleen of infected mice, as observed by increased numbers of CD11b cells and monocytes (CD11b$^+$Ly6C$^+$Ly6G$^-$) in BPTES-treated animals compared to untreated mice (Fig 2C). Interestingly, glutamine-supplemented mice displayed lower numbers of splenic monocytes (Fig 2C). The phenotypic characterization of the monocyte populations demonstrated an anti-inflammatory profile characterized by lower surface expression of M1-like markers including MHC-class II molecules and CD80 in BPTES-treated mice (Fig 2D). Furthermore, the numbers of IL-10-producing monocytes were increased in BPTES-treated mice when compared with untreated and glutamine-supplemented groups (Fig 2E). By contrast, we observed a reduction in the percentage of TNF-α-producing monocytes in BPTES-treated mice (Fig 2E). Our data also showed a significant increase in parasite burden in the spleen of BPTES-treated animals in comparison with their untreated counterparts (Fig 2F). This increase in parasite burden is likely a consequence of the increase number of splenic infected monocytes in BPTES-treated animals than to a direct effect of BPTES on parasite growth.

Altogether, our results show that *in vivo* GLS inhibition prompts an increased *L. donovani* burden through a mechanism that is related to the accumulation of monocytes displaying a suppressive profile.

### *In vivo* pharmacological inhibition of GLS impacts adaptive immunity against *L. donovani*

Given that BPTES-treated mice display an immunosuppressive phenotype, we then determined the *in vivo* impact on CD4 and CD8 T cells. To this end, we performed qualitative and quantitative analyses of T cells at day 12 post-infection. Our results indicated that BPTES-treated mice displayed lower numbers of both CD4 and CD8 T cells (Fig 3A) when compared to untreated mice. Herein, we observed that BPTES-treated mice displayed a reduced *Leishmania*-specific IFN-γ- and TNF-α-producing CD4 and TNF-α-producing CD8 T cells (Fig 3B). In parallel, there was a significant increase in the number of IL-10-producing CD4 T cells (Fig 3B). Previous data from our group have shown that increased IFN-γ/IL-10 ratios are associated with protection against *Leishmania* infection [18]. Furthermore, the increased *L. donovani* burden (Fig 2F) in BPTES-treated mice was associated with lower IFN-γ/IL-10 ratios of CD4 cells (Fig 3C). Interestingly, in mice supplemented with glutamine during *L. donovani* infection, splenic CD8 T cells expressed a tendency to increase IFN-γ and TNF-α levels and reducing the frequencies of CD8-expressing IL-10 in comparison with BPTES-treated mice (Fig 3B), suggesting that glutamine supplementation may improve CD8 T cell immunity. As such, an increase in IFN-γ/IL-10 ratio was observed in CD8 T cells of glutamine-supplemented animals compared to BPTES-treated mice (Fig 3C).

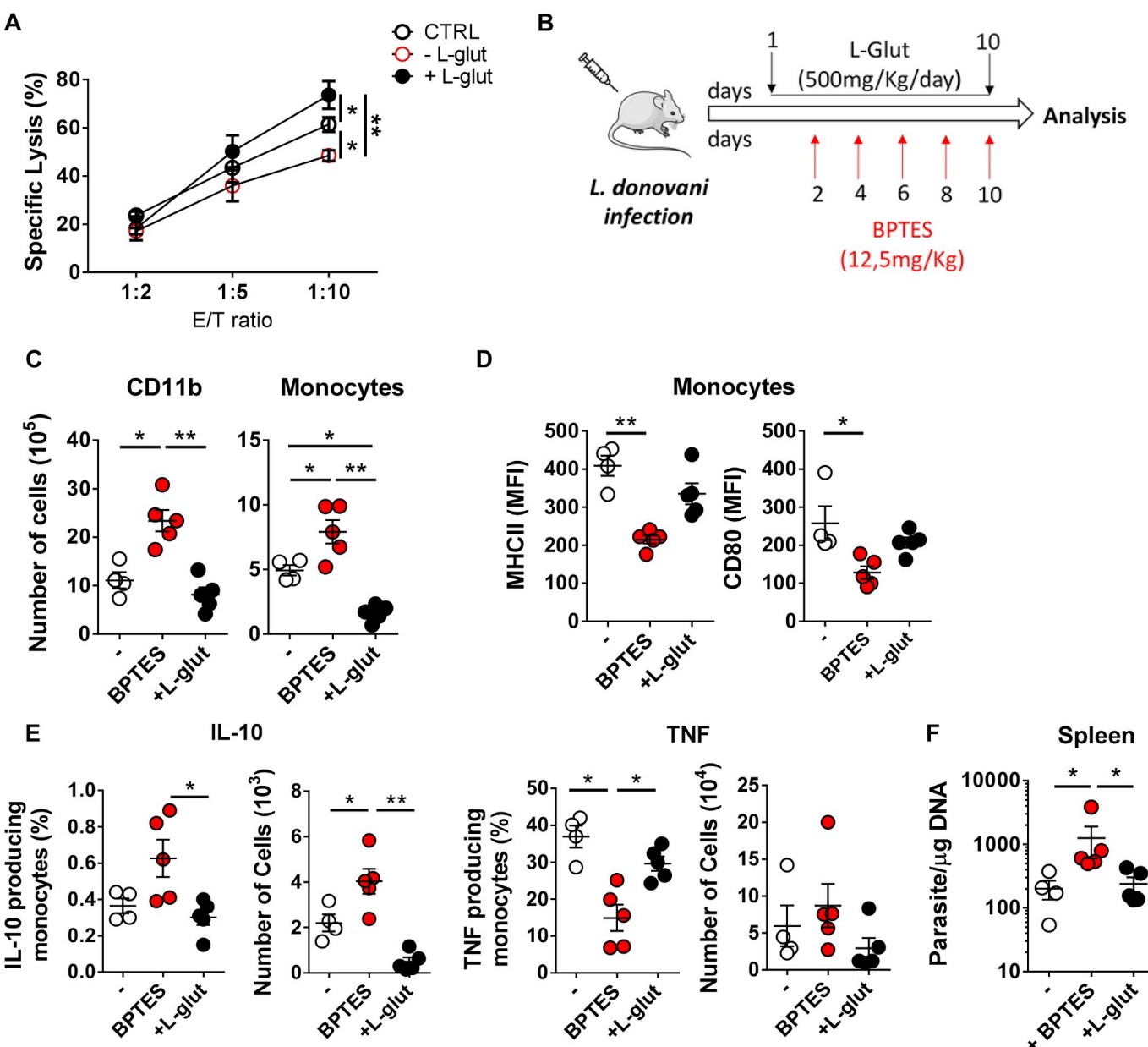

**Fig 2. GLS inhibition increases susceptibility of *L. donovani* infection.** (A) Spleen cells from 30 days infected mice were isolated and subjected to an *in vitro* stimulation with SLA (10 µg/ml) + IL-2 (20 ng/ml). After 3 days, the nonadherent cells were recovered and tested for *Leishmania*-specific cytotoxic activity in the presence of 2mM (CTRL), 5mM (+ L-glut) or absence (- L-glut) of glutamine. Twenty-four hours infected bone marrow derived macrophages were used as target cells. One way ANOVA test revealed significant differences at the E/T ratio 1:10 $F_{(2,16)}$ = 58,21, $p$ = 0.0014 (B) Mice were infected intraperitoneally with 1.0 x $10^8$ *L. donovani* promastigotes. BPTES was administered via intraperitoneal route at days 2, 4, 6, 8 and 10 post-infection (12.5 mg/kg). L-glutamine (500mg/Kg/day) was administered by oral gavage daily (starting at day 1 post-infection). (C) Kruskal-Wallis with Dunn's multiple comparison test found a significant main effects on the number of CD11b cells $F_{(3,12)}$ = 10,65, $p$ = 0.0001 and monocytes $F_{(3,12)}$ = 9,643, $p$ = 0.001 (CD11b$^+$ Ly6C$^+$ Ly6G$^-$) as quantified by flow cytometry in the spleen of all groups of mice. (D) Kruskal-Wallis with Dunn's multiple comparison test found a significant main effect on the surface expression of MHC-II $F_{(3,12)}$ = 10,28, $p$ = 0.0003 and CD80 $F_{(3,12)}$ = 8,523, $p$ = 0.0048 performed on gated monocytes between untreated and BPTES-treated groups. (E) Percentage and total number of IL-10 and TNF-α producing splenic monocytes were significantly different accordingly to Kruskal-Wallis with Dunn's multiple comparison test (Percentage of IL-10 $F_{(3,12)}$ = 6,174, $p<0.036$; IL-10 total numbers $F_{(3,12)}$ = 11,08, $p<0.0001$; percentage of TNF-α $F_{(3,12)}$ = 9,643, $p$ = 0,001) (F) Kruskal-Wallis with Dunn's multiple comparison test found a significant main effect in the parasite burden in the spleen of untreated, BPTES-treated and L-glutamine supplemented mice 12 days post-infection $F_{(3,12)}$ = 9,000, $p$ = 0,0032). Results shown in all panels are representative of 3 independent experiments. Data is shown as mean ± SD; n = 5 mice/group. Data is shown as mean ± SD; n = 5 mice/group. $^*p<0.05$; $^{**}p<0.01$; $^{***}p<0.001$.

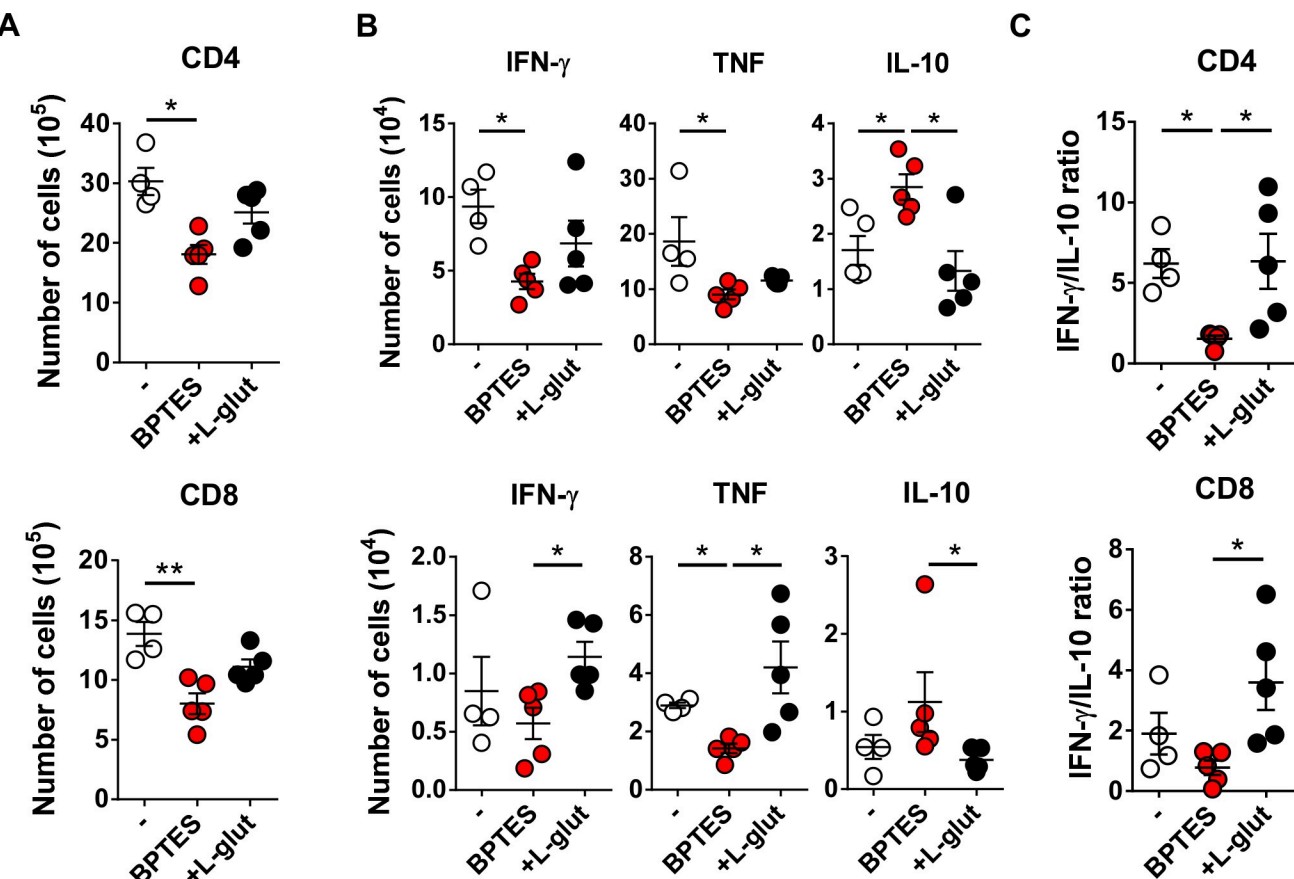

**Fig 3. Impairment of T cell *L. donovani* specific response associates with the increased susceptibility to infection upon GLS inhibition.** (A) At 12 days post-infection, Kruskal-Wallis with Dunn's multiple comparison test found a significant main effect in the total number of CD4 $F(3,12) = 8,381$, $p = 0,0055$) and CD8 T cells $F(3,12) = 10,05$, $p = 0,0004$) quantified in the spleen. (B) Intracellular production of IFN-γ, TNF-α and IL-10 was quantified in both CD4 and CD8 T cells upon 18 hours stimulation with SLA (10 ug/ml) and were significantly different accordingly to Kruskal-Wallis with Dunn's multiple comparison test (percentage of CD4+IFN-γ + $F(3,12) = 6,351$, $p<0.0323$; percentage of CD4+TNF-α+ $F(3,12) = 7,657$, $p<0.011$; percentage of CD4+IL-10+ $F(3,12) = 6,947$, $p<0.0217$; percentage of CD8+IFN-γ + $F(3,12) = 5,811$, $p<0.0464$; percentage of CD8+TNF-α+ $F(3,12) = 9,231$, $p<0.0019$; percentage of CD8+IL-10+ $F(3,12) = 7,586$, $p<0.0124$). (C) Kruskal-Wallis with Dunn's multiple comparison test found a significant main effect in the intracellular IFN-γ/IL-10 ratio determined for CD4 $F(3,12) = 9,000$, $p<0.0032$) and CD8 T cells $F(3,12) = 7,426$, $p<0.0148$). Results shown in all panels are representative of 3 independent experiments. Data is shown as mean ± SD; n = 5 mice/group. *$p<0.05$; **$p<0.01$; ***$p<0.001$.

These results demonstrated that the inhibition of glutaminolysis impaired effective protective T cell immune response increasing *L. donovani* parasite burden.

## Glutamine supplementation during *L donovani* infection improves treatment efficacy of miltefosine

Although the treatment of visceral leishmaniasis (VL) leads to a reduction of clinical signs, post-treatment relapses are common suggesting that parasites persist in the tissues. Thus, we have administrated miltefosine combined with a supplementation of glutamine (M + L-glut) in *L. donovani*-infected mice to test the potential of glutamine supplementation in the treatment of leishmaniasis. As control, we used miltefosine alone (Miltefosine) (Fig 4A). First, we quantified splenic T cells at day 32 post-infection. Whereas no significant difference was observed in the number of splenic CD4 T cells, we observed an increased number of CD8 T cells in M + L-glut-treated mice compared with untreated animals (Fig 4B). While miltefosine only increased the numbers of *Leishmania*-specific IFN-γ-producing CD4, the frequency of

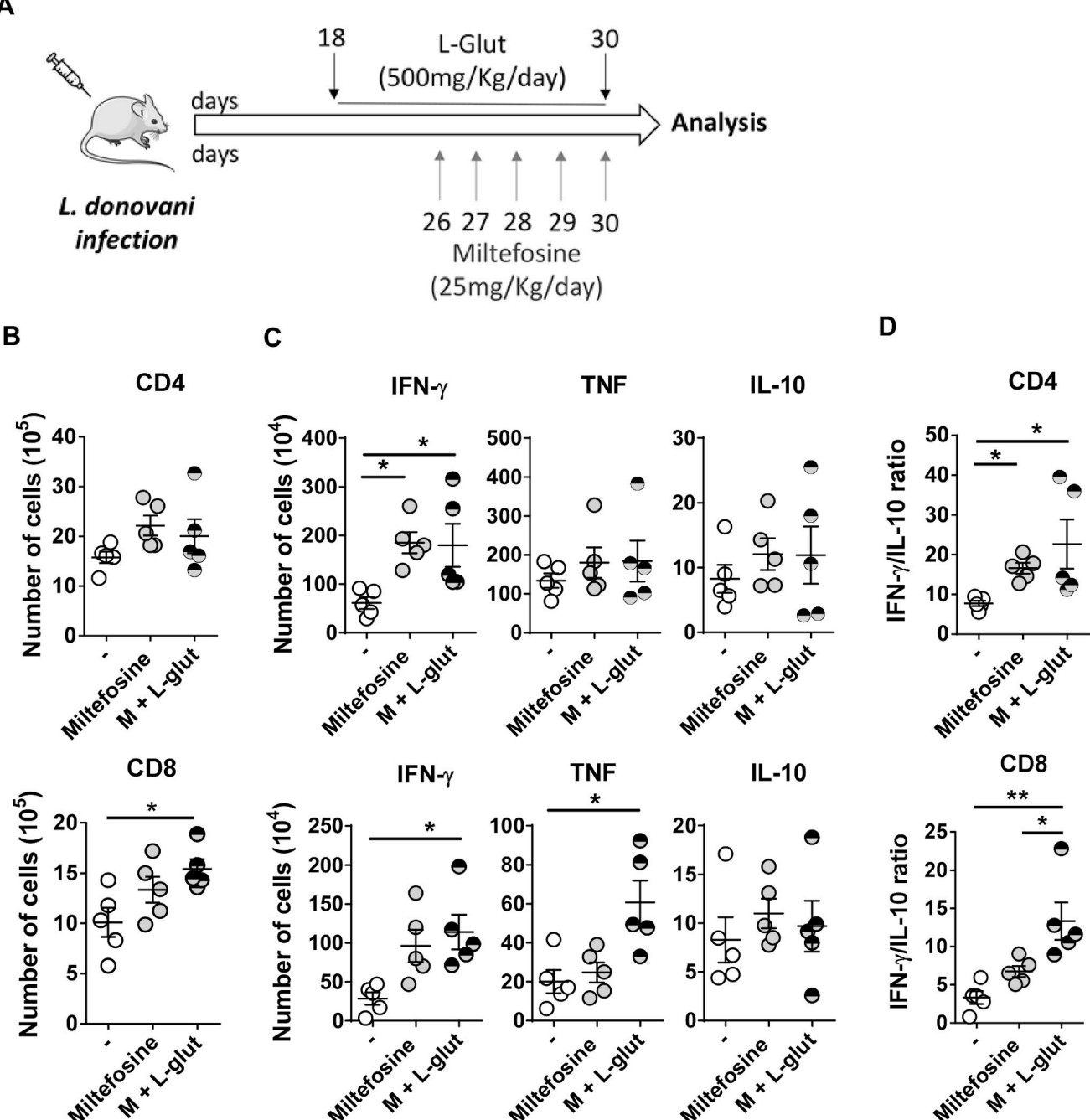

**Fig 4. L-glutamine supplementation to anti-*Leishmania* miltefosine treatment improves CD8 T cell inflammatory response.** (A) Mice were infected intraperitoneally with 1.0 x 10⁸ *L. donovani* promastigotes. L-glutamine (500 mg/kg/day) was administered by oral gavage daily (starting at day 18 post-infection). Miltefosine (25 mg/kg) was administered daily by oral gavage (starting at day 25 post-infection). Mice were euthanized one-month post-infection. (B) The total number of CD4 and CD8 T cells (significantly different accordingly to Kruskal-Wallis with Dunn's multiple comparison test F (3,12) = 5,805, *p*<0.0469) were quantified in the spleen. (C) Intracellular production of IFN-γ, TNF-α and IL-10 was quantified in both CD4 and CD8 T cells upon 18 hours stimulation with SLA (10 ug/ml). Kruskal-Wallis with Dunn's multiple comparison test found a significant main effect in the percentage of CD4⁺IFN-γ⁺ F(3,12) = 9,620, *p*<0.0018; percentage of CD8⁺IFN-γ⁺ F(3,12) = 9,302, *p*<0.0028; percentage of CD8⁺TNF-α⁺ F(3,12) = 8,060, *p*<0.0092) (D) Kruskal-Wallis with Dunn's multiple comparison test found a significant main effect in the intracellular IFN-γ/IL-10 ratio was determined for CD4 F(3,12) = 9,420, *p*<0.0024) and CD8 T cells F(3,12) = 11,06, *p*<0.0001). Results shown in all panels are representative of 3 independent experiments. Data is shown as mean ± SD; n = 5 mice/group. *p<0.05; **p<0.01; ***p<0.001.

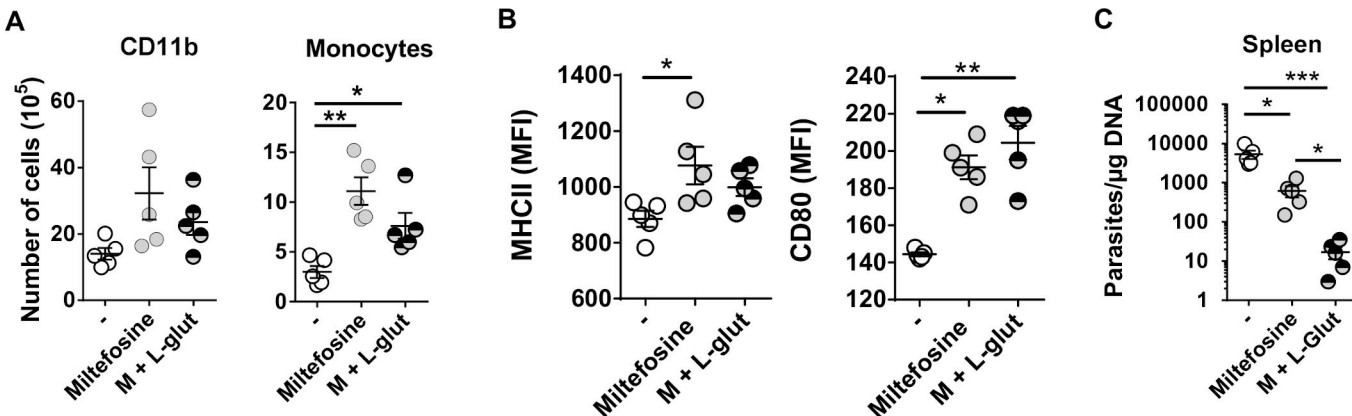

**Fig 5.** *In vivo* **glutamine supplementation increases the efficacy of miltefosine treatment during** *L. donovani* **infection.** (A) The number of CD11b cells and monocytes (significantly different accordingly to Kruskal-Wallis with Dunn's multiple comparison test F(3,12) = 11,18, $p < 0.0001$) were quantified by flow cytometry in the spleen of all groups of mice. (B) Kruskal-Wallis with Dunn's multiple comparison test found a significant main effect on the surface expression of MHC-II F(3,12) = 7,518, $p = 0.014$) and CD80 F(3,12) = 10,24, $p = 0.0006$) performed on CD11b⁺ Ly6c⁺ Ly6G⁻ (monocytes) gated cells. (C) Kruskal-Wallis with Dunn's multiple comparison test found a significant main effect in the parasite burden in the spleen of untreated (-), miltefosine-treated (Miltefosine) and L-glutamine-supplemented and miltefosine-treated animals (M + L-glut) F(3,12) = 12,50, $p < 0.0001$). Results shown in all panels are representative of 3 independent experiments. Data is shown as mean ± SD; n = 5 mice/group. *p<0.05; **p<0.01; ***p<0.001.

IFN-γ-producing CD4 and CD8 T cells and CD8-expressing TNF-α was improved in the adjunction of glutamine (Fig 4C). No significant impact was observed on the numbers of IL-10 producing T cells. Nevertheless, our results demonstrated that glutamine improve IFN-γ/IL-10 ratio on CD8 T cells (Fig 4D), which was previously associated with protection [18].

Both treatments increased splenic monocyte numbers compared to untreated animals without any significant impact on the CD11b⁺ population (Fig 5A). Additionally, the expression of MHC-II was significantly increased in monocytes from miltefosine-treated animals (Fig 5B) as well as the costimulatory molecule CD80 that was improved by the adjunction of glutamine (Fig 5B). Remarkably, L-glutamine supplementation in mice undergoing miltefosine treatment significantly decreased splenic parasite burden (Fig 5C), demonstrating that glutamine improves treatment efficacy of miltefosine through the development of a more effective anti-*Leishmania* adaptive immune response.

## Discussion

A growing body of evidence demonstrates that metabolic reprogramming of immune cells is extremely important for the development of an adequate immune response [3–5]. Immune cells use glutamine at similar or greater rates than glucose during several inflammatory states as sepsis, surgery recovery and high-intensity training [25–27]. Herein, we demonstrated an upregulation of the genes associated with glutaminolysis as well as an increase in glutamine consumption of *L. donovani*-infected macrophages. This is in agreement with a recent report by Koeken *et al* who showed an increased transcription of genes related to the glutamine pathway in *M. tuberculosis*-infected macrophages and patients with active tuberculosis [13]. Furthermore, modulation of glutaminolysis *in vivo* either by a specific GLS inhibitor or through glutamine supplementation impacts anti-*Leishmania* innate and adaptative immune responses. Remarkably, glutamine improves miltefosine treatment by enhancing effector T cells potentiating a reduction in the splenic parasite burden of *L. donovani*-infected mice.

Different strategies have been developed by parasites to persist in their infected hosts. Protozoa are able to suppress T cell-mediate immune response by limiting antigen presentation

by macrophages [28–30] or by altering T cell immunity allowing pathogen persistence [24]. Additionally, it is well established that the development of a protective adaptive immune response against *Leishmania* is characterized by *Leishmania*-specific cell-mediated cytotoxicity [31]. Herein, an improved cytotoxic capacity by anti-*Leishmania* specific T cells was observed by increasing the *in vitro* concentration of glutamine. This may be a consequence of increased antigen presentation by infected macrophages or a consequence of increased glutamine metabolism by T cells improving effector function, since glutamine metabolism is important for the activation status of both immune cell types [15,32]. In contrary, depletion of glutamine reduced the cytotoxic activity. In our co-culture system, it is plausible that a decreased bio-availability of glutamine is significantly hampering the cytotoxicity given that glutamine depletion blocks T cell proliferation and cytokine production [33,34]. Interestingly, *in vivo* administration of BPTES, a GLS inhibitor, led to a reduction on the surface expression of MHC class II and CD80 molecules, essential for antigen presentation and T cell co-stimulation [35,36], on myeloid cells recruited to the spleen than that observed in non-treated *L. donovani*-infected mice. Furthermore, GLS inhibition increased the recruitment of myeloid cells polarized with an anti-inflammatory profile, associated with a higher parasite burden. Herein, we observed that the glutamine pathway, although impact parasite growth in *in vitro* infected macrophages, rather contribute in rendering myeloid and T cells less efficient for performing immune function. Thus, glutamine modulation upon *L. donovani* infection represents an alternative way to hijack the immune functions for his own benefit.

Given that activated T cells upregulate amino acid transporters [14], glutamine metabolizing enzymes [15,37] as well as glutamine uptake [37,38], it is not surprisingly to observe that BPTES-treated mice display lower numbers of splenic CD4 and CD8 T cells, which pinpoints an essential role for glutamine metabolism during T cell activation in *L. donovani* infected mice. Our data also showed an important decrease in TNF-α and IFN-γ-producing CD4 and CD8 T cells. Previous reports have shown that GLS null T cells failed to drive Th17-inflammatory diseases contributing to Th1 exhaustion cells over time [34]. Additionally, glutamine deprivation or deletion of SLC1A5 was shown to promote Foxp3 expression, the transcription factor of regulatory T cells [15,39]. Xu and collaborators also suggested that inhibition or silencing of the enzyme responsible for the conversion of glutamate to α-ketoglutarate, GOT1, led to a shift in differentiation of Th17 to regulatory T cells [40]. We reported that such inhibition of glutamine pathway leads to an increased number of IL-10-producing lymphocytes and macrophages. This is consistent with a recent report showing that in a model of malnutrition, glutamine supplementation decreased the levels of IL-10 [41].

In nutrient restricted environments, as infection sites, there is a competition between host and pathogen for nutritional substrates due to their similar nutritional requirements (2). While the defensive response against pathogens is highly dependent on amino acid metabolism, *Leishmania* parasites are also able to modulate amino acid metabolism, and glutamine metabolism in particular, as a strategy to survive inside the host (3). Recent studies demonstrated that *Leishmania* amastigotes are highly dependent on mitochondrial metabolism for *de novo* synthesis of glutamate and glutamine [42]. *Leishmania* parasites, through glutamine synthetase (GS) action, convert glutamate into glutamine (5), playing a pivotal role in pyrimidine and hexosamine biosynthesis that are essential for parasite growth and stress responses [4,43]. The relevance of glutamine metabolism was demonstrated in studies where GS inhibition *in vitro* is detrimental for intracellular amastigote growth and virulence (6). Although these studies suggested that these stages are unable to salvage enough glutamate or glutamine from the macrophage phagolysosome [42], glutamine supplementation could revert it and instead provide an advantage for *Leishmania* survival and proliferation. Interestingly and in contrast with early glutamine supplementation, the additional intake of glutamine from day 18 to 30 post-

infection led to a tendency, although not significant, of increased parasite burden in the infected animals when compared with the untreated group (data not shown). Although our data demonstrate that glutamine supplementation *per se* is not a protective agent for parasite elimination, glutamine supplementation as an adjuvant to Miltefosine potentiated its anti-*Leishmania* activity. Given that miltefosine is not only capable of inducing direct parasite killing but also of modulating the host immunity by activating Th1 cytokines particularly represented by increased IFN-γ and interleukin 12 (IL-12) [44] and this activation is absolutely dependent on extracellular glutamine consumption [33], we hypothesized that the increased glutamine bioavailability will be crucial for the fitness of T cell response potentiating the development of a more effective anti-leishmania immune response.

To date, miltefosine is the only recognized oral drug used for visceral leishmaniasis treatment. However, miltefosine resistance has been recently demonstrated in the Indian subcontinent due to the increasing evidence of post-treatment relapses [45]. Thus, despite the reduction of the clinical symptoms, the actual therapy does not completely eradicate the parasites in infected individuals. Glutamine supplementation is nowadays routinely used in several applications, for example for pre-and post-operative patients to restore immune function [46]. We demonstrated that *in vivo* glutamine supplementation boosts CD4 and CD8 T cells improving IFN-γ and TNF-α expression, particularly in CD8 T cells, in mice infected with *L. donovani*. Of interest, our results also highlighted that monocytes recruited in the spleen showed an increased expression of CD80. Remarkably, glutamine supplementation during miltefosine treatment displays a synergistic effect for the treatment of visceral leishmaniasis. Thus, glutamine supplementation during *L. donovani* infection improves immune response by drastically reducing the parasite burden.

In conclusion, our study has identified glutaminolysis as a fundamental metabolic process essential for the *in vivo* control of *L. donovani*. More importantly, dietary glutamine supplementation may improve the efficacy of miltefosine treatment and might be considered as an adjuvant for visceral leishmaniasis treatment.

## Supporting information

**S1 Table. List of anti-mouse monoclonal antibodies.**
(DOCX)

**S2 Table. List of primers used in qRT-PCR.**
(DOCX)

## Author Contributions

**Conceptualization:** Carolina Ferreira, Bhaskar Saha, Jerôme Estaquier, Ricardo Silvestre.

**Data curation:** Carolina Ferreira, Nuno Sampaio Osório, Charles-Joly Beauparlant, Arnaud Droit, Ricardo Silvestre.

**Formal analysis:** Carolina Ferreira, Nuno Sampaio Osório, Charles-Joly Beauparlant, Arnaud Droit.

**Investigation:** Carolina Ferreira, Inês Mesquita, Ana Margarida Barbosa.

**Methodology:** Carolina Ferreira, Inês Mesquita, Ana Margarida Barbosa, Ricardo Silvestre.

**Project administration:** Ricardo Silvestre.

**Resources:** Egídio Torrado, Cristina Cunha, Agostinho Carvalho, Jerôme Estaquier.

**Supervision:** Jerôme Estaquier, Ricardo Silvestre.

**Validation:** Jerôme Estaquier.

**Visualization:** Carolina Ferreira, Ricardo Silvestre.

**Writing – original draft:** Carolina Ferreira.

**Writing – review & editing:** Inês Mesquita, Nuno Sampaio Osório, Egídio Torrado, Cristina Cunha, Agostinho Carvalho, Bhaskar Saha, Jerôme Estaquier, Ricardo Silvestre.

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
