## [Decision Letter · Decision Letter 0]

17 Dec 2019

Dear Dr Silvestre:

Thank you very much for submitting your manuscript "Glutamine supplementation improves the efficacy of miltefosine treatment for visceral leishmaniasis" (#PNTD-D-19-01729) for review by PLOS Neglected Tropical Diseases. Your manuscript was fully evaluated at the editorial level and by independent peer reviewers. The reviewers appreciated the attention to an important problem, but raised some substantial concerns about the manuscript as it currently stands. These issues must be addressed before we would be willing to consider a revised version of your study. We cannot, of course, promise publication at that time.

We therefore ask you to modify the manuscript according to the review recommendations before we can consider your manuscript for acceptance. Your revisions should address the specific points made by each reviewer. 

When you are ready to resubmit, please be prepared to upload the following:

(1) A letter containing a detailed list of your responses to the review comments and a description of the changes you have made in the manuscript.

(2) Two versions of the manuscript: one with either highlights or tracked changes denoting where the text has been changed (uploaded as a "Revised Article with Changes Highlighted" file); the other a clean version (uploaded as the article file).

(3) If available, a striking still image (a new image if one is available or an existing one from within your manuscript). If your manuscript is accepted for publication, this image may be featured on our website. Images should ideally be high resolution, eye-catching, single panel images; where one is available, please use 'add file' at the time of resubmission and select 'striking image' as the file type. 

Please provide a short caption, including credits, uploaded as a separate "Other" file. If your image is from someone other than yourself, please ensure that the artist has read and agreed to the terms and conditions of the Creative Commons Attribution License at http://journals.plos.org/plosntds/s/content-license (NOTE: we cannot publish copyrighted images). 

(4) If applicable, we encourage you to add a list of accession numbers/ID numbers for genes and proteins mentioned in the text (these should be listed as a paragraph at the end of the manuscript). You can supply accession numbers for any database, so long as the database is publicly accessible and stable. Examples include LocusLink and SwissProt.

(5) To enhance the reproducibility of your results, we recommend that you deposit your laboratory protocols in protocols.io, where a protocol can be assigned its own identifier (DOI) such that it can be cited independently in the future. For instructions see http://journals.plos.org/plosntds/s/submission-guidelines#loc-methods

While revising your submission, please upload your figure files to the Preflight Analysis and Conversion Engine (PACE) digital diagnostic tool, https://pacev2.apexcovantage.com/ PACE helps ensure that figures meet PLOS requirements. To use PACE, you must first register as a user. Then, login and navigate to the UPLOAD tab, where you will find detailed instructions on how to use the tool. If you encounter any issues or have any questions when using PACE, please email us at figures@plos.org.

We hope to receive your revised manuscript by Feb 15 2020 11:59PM. If you anticipate any delay in its return, we ask that you let us know the expected resubmission date by replying to this email.

To submit a revision, go to https://www.editorialmanager.com/pntd/ and log in as an Author. You will see a menu item call Submission Needing Revision. You will find your submission record there. 

Sincerely,

Peter C. Melby, M.D.

Associate Editor

Charles Jaffe

Deputy Editor

Reviewer's Responses to Questions

**Key Review Criteria Required for Acceptance?**

**Methods**

-Are the objectives of the study clearly articulated with a clear testable hypothesis stated?

-Is the study design appropriate to address the stated objectives?

-Is the population clearly described and appropriate for the hypothesis being tested?

-Is the sample size sufficient to ensure adequate power to address the hypothesis being tested?

-Were correct statistical analysis used to support conclusions?

-Are there concerns about ethical or regulatory requirements being met?

Reviewer #1: See "Summary and General Comments"

Reviewer #2: The objectives of the study are clearly articulated with the hypothesis stated, the study is well designed and the ethical requirements are being met. As suggested in the major comments for the authors, the metodology for transcriptome analysis must be better detailed. On this regard, information is missing.

**Results**

-Does the analysis presented match the analysis plan?

-Are the results clearly and completely presented?

-Are the figures (Tables, Images) of sufficient quality for clarity?

Reviewer #1: See "Summary and General Comments"

Reviewer #2: Data are well presented, results are clear and match the analysis plan. Figures are also clear.

**Conclusions**

-Are the conclusions supported by the data presented?

-Are the limitations of analysis clearly described?

-Do the authors discuss how these data can be helpful to advance our understanding of the topic under study?

-Is public health relevance addressed?

Reviewer #1: See "Summary and General Comments"

Reviewer #2: The conclusions are supported by the data presented. However, it is suggested to moderate some of them in order to avoid reductionisms. Limitations of analysis were not described.

**Editorial and Data Presentation Modifications?**

Reviewer #1: See "Summary and General Comments"

Reviewer #2: (No Response)

**Summary and General Comments**

Reviewer #1: The study by Ferreira and collaborators provides interesting data indicating that dietary glutamine supplementation may act as a promising adjuvant for the treatment of visceral leishmaniasis. The study is well conducted, but I have some concerns, especially regarding analytical (statistical) procedures. Here are some additional comments.

1) 1) Is there any reason for choosing C57Bl/6J mice? Wouldn't Balb/c mice have been a more susceptible Leishmania donovani?

2) Was the in vitro cytotoxicity assay performed from any previous work? If yes, cite the reference.

3) Line 166: The route of exposure was intraperitoneal rather than intravenous. What is the reason for this? This detail should be clarified in the manuscript.

4) Line 171: The criteria for choosing the dose administered is unclear.

5) What is the sample number of animals per group?

6) Tables 1 and 2 may be reallocated to “supplementary material”.

7) Statistical analyzes are incorrect. Data should be analyzed using a factorial ANOVA. There are different factors which should be considered (gender, time, treatments). Therefore, the authors must redo the analyzes and present the results appropriately (effect of the factors separately and the interaction between them - present F values, degrees of freedom and p values). In addition, ANOVA assumptions must be properly tested.

Reviewer #2: Comment to PNTD-D-19-01729

The manuscript reports on the modulation of glutamine metabolism, in the vertebrate host, by Leishmania donovani and suggests a potential therapeutic use of glutamine in combination with miltefosine for the treatment of visceral leishmaniasis. The modulation of the metabolism of the host cells by these parasites is a relevant issue for the understanding of the immune mechanisms that control the infection. However, this topic is little studied. In this sense, the manuscript addresses an interesting topic and provides important new insights into the crosstalk between glutamine metabolism and T cell-mediated immune response to L. donovani infection. Overall, the manuscript is very well written and the data are well presented. Conclusion could be more cautious as suggested below. 

Major comments:

Lines 181-187. Detailed information on the transcriptomic analysis is lacking. Information about whether it was a time course analysis, BM or peritoneal macrophages? Number of experiments? RNA Information, such as it was total or poly-A RNA, if it was treated for depletion of ribosomal DNA, how it was isolated and quantified, which quality control was used. How library was constructed? This information must be included for the correct interpretation of the data by the reader. 

In addition, It is not clear whether transcriptomic assays were done at 6, 24 and 48 hours or if they were done just at 6h post-infection and the other time points refer to other assays of gene expression analysis. The design of transcriptomic analysis must be detailed.

Lines 286-287. It would be probably because those cells stopped proliferating due to the lack of glutamine? Please elaborate on that. 

Lines 297-298. How do authors explain that there was no difference between the untreated and the group supplemented with glutamine? Shouldn't there be an improvement in animals that received supplementation compared to untreated ones?

Lines 318-320 and figure 3. The supplemented group also decreased the number of CD4 and CD8 T cells. Authors expected that? How to explain this? Shouldn't it be at least the same as the untreated? Or even increase thanks to supplementation? Also, how to explain the reduction of IFN-γ- and TNF produced by CD4 T cells? The authors are encouraged to elaborate on this result. 

For this reviewer, the conclusions need to be more moderate because, as it is written, it seems to reduce the complexity the parasite-host relationship. It would be ideal if supplementation with glutamine was sufficient to contribute for controlling the infection, but that would be reductionist, since many other factors contribute to the control of the parasite or, conversely, to the establishment of the infection. In fact, parasites also consume glutamine to maintain their proliferation, and a medium supplemented in this amino acid could also result in parasite survival and establishment within the cells. This aspect deserves to be mentioned by the authors. Could parasites compete with macrophages for glutamine? Please elaborate on that and moderate conclusions. 

Minor points:

Figure 3. Captions of this figure should be standardized, i.e., use the same font size.

PLOS authors have the option to publish the peer review history of their article (what does this mean?). If published, this will include your full peer review and any attached files.

Reviewer #1: Yes: Prof. Dr. Guilherme Malafaia

Reviewer #2: No

---

## [Editor Report · Decision Letter 1]

9 Feb 2020

Dear Silvestre,

We are pleased to inform you that your manuscript 'Glutamine supplementation improves the efficacy of miltefosine treatment for visceral leishmaniasis' has been provisionally accepted for publication in PLOS Neglected Tropical Diseases.

Before your manuscript can be formally accepted you will need to complete some formatting changes, which you will receive in a follow up email. A member of our team will be in touch within two working days with a set of requests.

Best regards,

Peter C. Melby, M.D.

Associate Editor

Charles Jaffe

Deputy Editor

The authors have adequately addressed the concerns of the reviewers. The work is a nice contribution to the field. I have 2 minor editorial comments related to the legend to figure 1: Please identify the method used for determination of gene expression in Figure 1A and Figure 1B. The sentence in line 660-61 ("A two-way ANOVA. . .) is out-of-place.

---

## [Editor Report · Acceptance letter]

11 Mar 2020

Dear Silvestre,

We are delighted to inform you that your manuscript, "Glutamine supplementation improves the efficacy of miltefosine treatment for visceral leishmaniasis," has been formally accepted for publication in PLOS Neglected Tropical Diseases.

Best regards,

Serap Aksoy

Editor-in-Chief

Shaden Kamhawi

Editor-in-Chief
